# Plant Taxonomic Diversity Better Explains Soil Fungal and Bacterial Diversity than Functional Diversity in Restored Forest Ecosystems

**DOI:** 10.3390/plants8110479

**Published:** 2019-11-06

**Authors:** Md. Abu Hanif, Zhiming Guo, M. Moniruzzaman, Dan He, Qingshui Yu, Xingquan Rao, Suping Liu, Xiangping Tan, Weijun Shen

**Affiliations:** 1Key Laboratory of Vegetation Restoration and Management of Degraded Ecosystem, South China Botanical Garden (SCBG), Chinese Academy of Sciences, Guangzhou 510650, Chinaguozhiming_ecology@163.com (Z.G.); danhe@scbg.ac.cn (D.H.); yuqingshui@pku.edu.cn (Q.Y.); rxq99@scbg.ac.cn (X.R.); spliu@scbg.ac.cn (S.L.); tanxp@scbg.ac.cn (X.T.); 2University of Chinese Academy of Sciences, Beijing 100000, China; 3Department of Agroforestry and Environment, Hajee Mohammad Danesh Science and Technology University, Dinajpur-5200, Bangladesh; 4Institute of Fruit Tree Research, Guangdong Academy of Agricultural Sciences, Guangzhou 510640, China; monirbge@gmail.com

**Keywords:** plant-soil feedback, soil bacterial community, soil fungal community, taxonomic diversity, plant functional traits, 16S sequencing

## Abstract

Plant attributes have direct and indirect effects on soil microbes via plant inputs and plant-mediated soil changes. However, whether plant taxonomic and functional diversities can explain the soil microbial diversity of restored forest ecosystems remains elusive. Here, we tested the linkage between plant attributes and soil microbial communities in four restored forests (*Acacia* species, *Eucalyptus* species, mixed coniferous species, mixed native species). The trait-based approaches were applied for plant properties and high-throughput Illumina sequencing was applied for fungal and bacterial diversity. The total number of soil microbial operational taxonomic units (OTUs) varied among the four forests. The highest richness of fungal OTUs was found in the *Acacia* forest. However, bacterial OTUs were highest in the *Eucalyptus* forest. Species richness was positively and significantly related to fungal and bacterial richness. Plant taxonomic diversity (species richness and species diversity) explained more of the soil microbial diversity than the functional diversity and soil properties. Prediction of fungal richness was better than that of bacterial richness. In addition, root traits explained more variation than the leaf traits. Overall, plant taxonomic diversity played a more important role than plant functional diversity and soil properties in shaping the soil microbial diversity of the four forests.

## 1. Introduction

Biodiversity conservation and protection of land from degradation are the key strategies behind ecosystem restoration [1]. Half of the world’s degraded tropical forests are restored through reforestation or are converted to secondary plantation forests [2]. These degraded forests are restored with monoculture or mixed plantations as a process of the restoration strategy [3]. However, the success of ecological restoration largely depends upon the interactions between above and belowground communities driving ecosystem processes [1,4]. We lack empirical evidence on plant-microbe linkage from restored forest landscapes, especially those exploring the relative contribution of plant attributes and soil properties to explain soil microbial diversity.

Plantation types, preference of habitats and quality of plant inputs are the key factors that influence soil microbial communities during restoration [5,6]. Vegetation types having contrasting diversity of plant communities will have a distinct effect on soil properties through plant inputs, which in turn has diverse effects on soil microbial communities [3,4]. Moreover, the individual species alter soil chemical properties through the species-specific chemistry of litter inputs, which inevitably affects soil microbial communities [7,8]. The presence of a higher diversity of productive species can also have a strong influence on soil functions and microbial diversity [9,10]. For example, long-term restoration with *Pinus massoniana* and *Eucalyptus* spp. causes soil degradation, which profoundly influences soil microbial communities [4]. In *Pinus elliottii* plantations, fungal biomass increased, but bacterial biomass decreased due to microecological imbalance and a gradual decrease in the quality of inputs during long-term restoration in subtropical China [3]. Specific species were also found to have a contrasting impact on specific taxa of bacteria during forest conversion from native to teak plantations [11]. Therefore, how soil fungal and bacterial communities respond to vegetation restoration with respect to different plantations needs to be explored. We hypothesized that different plantations in our study restored with contrasting species will vary in terms of plant richness and diversity and the distinct effect of plant-mediated soil changes will influence soil fungal and bacterial communities.

Plant taxonomic and functional diversity affect soil microbial diversity during restoration [5,12] by altering the available resources [13,14], niche differentiation and resource partitioning [15,16]. In natural ecosystems, soil fungal diversity is inevitably dependent on species richness indicating the importance of individual species that generate complementary belowground niches by their inputs [17,18,19]. Roy-bolduc et al. [20] found a strong positive relationship between plant diversity and soil fungal diversity. In contrast, Shi et al. [21] reported an inverse relationship between tree diversity and fungal diversity. Bacterial diversity was also found to decrease with the species richness along latitudinal gradients where environmental conditions act as the predominant driver [13]. Distinct bacterial groups were found in broadleaf forest and coniferous forest during restoration in the mountain region of China [5]. Therefore, plant taxonomic diversity will have an impact on soil microbial diversity, but to better understand the effect of plant attributes on the soil microbial community, it is necessary to determine the relative contribution of plant functional diversity along with plant taxonomic diversity under the vegetation change regime [12].Community-weighted mean (CWM) traits of dominant species and multi-trait functional dispersion (FD) are the two important hypotheses by which we can find the effect of plant functional diversity on soil microbial diversity. Ecosystem functions and properties are intensely influenced by the traits of the dominant species [22,23]. Based on the biomass ratio hypothesis (CWM traits) [24], attributes of dominant species in a community regulate ecosystem properties, and the quality and quantity of litter traits of dominant species play a key role in regulating soil microbial richness [25]. Moreover, based on the niche complementarity hypothesis, FD also plays a pivotal role in several ecosystem functions [26]. FD can facilitate niche partitioning, which creates more available resources for niche spaces and is beneficial for microbial communities [16]. Therefore, it is imperative to include both CWM and FD to better understand the effect of plant functional trait diversity on microbial communities. We hypothesized that under this circumstance, the proportional contribution of plant taxonomic diversity will be greater than that of the functional diversity in explaining soil microbial diversity.

In recent years, trait-based approaches have been applied for the prediction of fungal and bacterial diversities at the individual plant [9], community [27], and regional scales [28]. The richness of some groups of soil microbes, such as mycorrhizal fungi and archaeal ammonia oxidizers, is influenced by plant functional traits [29,30]. Plant functional traits such as specific leaf area (SLA) and leaf nitrogen concentrations (LN) can alter soil properties via the input of litter and detritus, which can affect the richness of soil microbes [31,32]. Moreover, the diversity and richness of soil microbial communities are affected by interspecific variation in both the quality and quantity of resource inputs [33,34,35]. Microbial properties were found to have a closer linkage with belowground root traits than leaf traits in the grasslands of Europe [27]. Root nutrients such as root nitrogen concentration (RN) and the root carbon: nitrogen ratio (C: N) were found to have a direct effect, whereas leaf traits (SLA, shoot N, and C: N) indirectly influence soil fungal and bacterial diversity [31]. Previous studies have mostly included only aboveground plant functional traits [25,28,32] or, in some instances, a few belowground traits to describe the plant-microbe interaction [27]. However, the combination of both aboveground and belowground traits and their proportional contribution in explaining soil microbial diversity would be a more viable option. Therefore, we hypothesized that the root traits might contribute more than the leaf traits in explaining belowground soil microbial diversity.

Here, we explored the role of plant diversity (taxonomic and functional diversity) and plant functional traits (aboveground and belowground) as predictors of belowground soil fungal and bacterial diversities in four subtropical plantation forests of southern China. The four types of forests included an *Acacia mangium*(AM) forest, a mixed *Eucalyptus* species (EE) forest, a mixed coniferous species forest (MC) of *Cunninghamia lanceolata* and *Pinus massoniana,* and a mixed *Schima* species (NS) forest. Specifically, we aimed to address three questions: (1) How does the soil microbial (fungi and bacteria) community structure differ among the four plantations? (2) What is the relative importance of plant functional diversity vs. taxonomic diversity for microbial diversity? (3) Which plant traits (aboveground or belowground) contribute more to determining the soil microbial community composition? We hypothesized that (1) restoration with diverse species through plant-mediated soil changes might alter soil microbial diversity [5,6,7]; (2) taxonomic diversity might contribute more than functional diversity due to species-specific linkage between microbes and individual plants [17,18,19]; and (3) belowground traits might contribute more due to their close association with soil microbes [27].

## 2. Results

### 2.1. Plant and Soil Community Composition

Species richness in the four forests ranged from 7 to 20 (Table A1), and species richness among the four forests was found to be nonsignificant (Figure 1a). Plant diversity (Shannon index) differed among the forests and ranged from 1.035 to 2.546 (Figure 1b). Maximum plant diversity was found in the EE forest, which was statistically similar to that in the MC and EE forests, while minimum plant diversity was recorded in the AM forest (Figure 1b).

Fungal operational taxonomic unit(OTU) richness and diversity differed among the four forests (Table A2) and ranged from 91 to 471 (Figure 2a) and 1.202 to 6.168 (Figure 2c). The maximum fungal OTU richness and diversity were recorded in the AM forest (Figure 2a,c). The fungal OTU richness in the MC forest was statistically similar to that of the AM forest (Figure 2a). Minimum fungal OTU richness and diversity were found in the NS forest (Figure 2a,c). Bacterial OTU richness and diversity in the four forests also varied (Table A3) and ranged from 561 to 659 (Figure 2b) and 7.35 to 7.92 (Figure 2d). The maximum bacterial OTU richness and diversity were found in the EE forest (Figure 2b,d), which were statistically similar to the MC and NS forests, while the minimum bacterial OTU richness and diversity were recorded in the AM forest (Figure 2b,d).

The results of ANCOVA analysis revealed that the fungal richness and diversity differed significantly (*P* < 0.05) in terms of forest types (Table A3). In the case of bacteria, forest types significantly influenced bacterial richness (*P* < 0.05) but not bacterial diversity (*P* > 0.05). Fungal and bacterial OTU richness were found to be significantly related to the abundance of common and dominant species across the four forests (Table A4). The abundance of *Illex asprella* was found to have a positive significant relationship with fungal richness, while bacterial richness was positively and significantly related to the abundance of *Melicope pteleifolia* and *Gardenia jasminoides*(Table A4).

According to the ITS sequence reads, the fungal community (Figure A1) was mainly composed of Basidiomycota (47.8%), Ascomycota (32.4%), and Zygomycota (13.4%), whereas bacterial phyla (Figure A2) were composed of Acidobacteria (45.7%), Proteobacteria (28.0%), and Chloroflexi (7.7%).

Soil properties across the four forests varied significantly. The total soil carbon (SOC) and total soil nitrogen (TN) were significantly higher in the AM forest, while other forests had significantly similar amounts of SOC and TN (Table A5). The total soil phosphorus (TP) was found to be significantly highest in NS forest; total soil potassium (TK) was found highest in MC, while TP and TK were found to be significantly lower in the EE forest (Table A5). Acidic soil was recorded in all forests with pH ranged from 3.76 to 3.95. Soil moisture did not vary significantly among the four forests and the range was between 24.58% to 32.62% (Table A5). Pearson correlation analysis found SOC and TN were significantly related either with fungal/bacterial richness and diversity across the four forests (Table A6).

### 2.2. Relative Contribution of Plant Taxonomic Diversity Versus Functional Diversity in Explaining Fungal and Bacterial Diversity

Plant species richness significantly and positively described 36% of the variation in fungal richness and 56% of the variation in bacterial richness (Figure 3a,b).

The redundancy analysis (RDA) results revealed that 93% of the variation in fungal richness and diversity was explained by plant attributes and soil properties (Figure 4a); species richness (PSPRICH), leaf dry matter content (LDMC), leaf phosphorus content (LP), specific root length (SRL), root dry matter content (RDMC), SOC and TN were the significant variables. In the case of bacteria, 78% of the variation was explained by the predictors (plant attributes and soil properties), and the significant variables were PSPRICH, FD, leaf carbon content (LC), LP, leaf vein density (VD), root carbon content (RC) and root phosphorus content (RP) (Figure 4b).

Furthermore, the results from the ANCOVA revealed that the contributions of plant attributes differed in explaining the fungal and bacterial richness and diversity (Table A3). Fungal richness was significantly influenced by species richness and CWM traits, while fungal diversity was significantly influenced by both CWM traits and FD along with plant diversity (Table A3). Bacterial richness was significantly influenced by species richness and FD, while bacterial diversity was significantly influenced by plant diversity along with FD.

Variance partitioning analysis (VPA) revealed that plant taxonomic diversity (27%), functional diversity (4%), and soil properties (3%) explained 34% (individual effect) of the variation in fungal communities. These three predictors, including their interaction effect, explained 46% of the variation of soil fungi dwelling in bulk soil (Figure 5). For bacteria, plant taxonomic diversity and soil properties explained 22% of the variation, with the individual contribution from plant taxonomic diversity being 19% and that from soil properties being 3%. Plant functional diversity alone had little explanatory power for bacterial communities; however, together with plant taxonomic diversity, it predicted 18% of the richness and diversity of bacterial communities (Figure 5).

### 2.3. Functional Traits Variation across Forest Ecosystems and Contribution of Functional Traits for Explaining Microbial Diversity

The CWM of leaf (SLA, LDMC, VD, LN, LP) and root (SRL, RDMC, RN, RC) traits varied significantly (*P* < 0.05) among the four forests (Table 1). The highest CWM of leaf (SLA, LDMC, VD, LP) and root (RDMC, RC, RN) traits were found in the AM forest. LN was highest in the EE forest; and the maximum SRL was recorded in the NS forest. The lowest LDMC and RDMC were found in the EE forest; the lowest SRL was found in the MC forest; and the minimum values of SLA, VD, LN, LP, and RN were recorded in the NS forest (Table 1). The CWM of leaf and root C: N varied significantly among the four forests; the maximum C: N for both roots and leaves were recorded in the NS forest, and the minimum was recorded in the EE forest (Figure A3).

Single trait functional dispersion (Table A7) and multi-trait FD also varied among the four forests; FD was significantly higher in the AM and NS forests than in the two other forests (Table 1). The single trait functional diversity of leaf traits was significantly maximum in the MC forest, which was statistically similar to the AM forest, while the minimum was found in NS and EE forests (Table A7). In the case of root traits, the maximum value was recorded in the NS forest, which was statistically similar to the AM forest, and the minimum was found in the EE and MC forests (Table A7).

We used the multimodal inference to explain belowground microbial richness. We found that the overall prediction for fungi (R^2^ = 0.76, *P* < 0.001) was better than that for bacteria (R^2^ = 0.45, *P* < 0.001) (Table 2). Functional traits (both aboveground and belowground) better explained fungal richness (R^2^ = 0.68, *P* < 0.001) than bacterial richness (R^2^ = 0.31, *P* < 0.011). Height, LDMC, VD, LN, and LP were the best aboveground traits (R^2^ = 0.42, *P* < 0.001) for explaining fungal richness, whereas for bacteria, LDMC, LC, LN, and LP were the best predictors (R^2^ = 0.16, *P* < 0.032). For the belowground traits, SRL, RDMC, RC, and RP (R^2^ = 0.43, *P* < 0.001) were the best predictors for fungi, and RDMC, RC, RN, and RP (R^2^ = 0.22, *P* < 0.036) better explained bacterial richness. Belowground traits explained more than the aboveground traits for both fungi and bacteria (Table 2).

## 3. Discussion

Forest restoration with different plant species influenced soil microbial diversity. Fungal richness and diversity were highest in the AM plantation. AM is the leguminous forest with abundant *Acacia mangium* species. Restoration with leguminous plants (*Acacia*) fixes atmospheric nitrogen, which is added to the soil and might influence higher fungal communities in AM forests [36,37]. Moreover, SOC and TN (Table A5) were significantly higher in the AM forest than in the other forests, which perhaps influences the soil fungal community. Among the edaphic factors, soil fertility (for example, SOC, TN) is the key edaphic factor that influences soil microbial richness [13] because SOC and TN provide energy to soil fungi [6], which increases their activity and subsequently increases fungal diversity [7,8]. The CWM traits of the leaves (SLA, LC, and LP) and roots (RDMC, RC, and RN) were higher in AM forest (Table 1) suggesting the presence of more exploitative species that stimulate rapid acquisition and turnover, thus facilitating fungal composition [31,38]. Moreover, the multi-trait FD and single trait functional diversity measures were highest in the AM forest, which indicates that the higher resource availability (nutrients entering the soil via the plant parts) was present in this site, leading to more availability of niche space for fungi [39,40,41,42]. Therefore, vegetation restoration with leguminous plants compared to other plants might facilitate the activity of the fungal community due to increased resource availability [43].

Bacterial richness and diversity were lower in AM forest. The quality of nutrients and lower plant diversity in the AM forest compared to the other forests might be attributed to the lower bacterial richness and diversity. In contrast, bacterial richness was highest in EE plantation where the plant diversity was also found highest among the studied forests. The higher soil bacterial richness in the EE forest might be influenced by the quality of the substrate entering the soil [44]. Leaf and root C: N ratios were significantly lower in the EE forest than in the other forests (Figure A3); therefore, the quality of the inputs was higher in the EE forest, which might facilitate higher bacterial richness [13]. Moreover, bacterial richness may be more associated with the abundance of specific plants (e.g., *Melicope pteleifolia*, *Gardenia jasminoides*) in the EE forest.

Plant taxonomic diversity explained more of the soil microbial diversity than the functional diversity and soil properties, which supports our second hypothesis. Soil microbial (fungi and bacteria) richness increased with species richness. This might be due to the greater diversity of organic substrates, resources and carbon compounds for soil microbes [40,43,45]. The VPA analysis also indicated that plant taxonomic diversity better explained fungal diversity than bacterial diversity (Figure 5). This result indicates that the individual tree effect is much stronger in shaping fungal and bacterial diversity [5,6,12,40]. Fungal richness was found to increase with the increase of the abundance of some specific species (*Illex asprella* and *Clerodendrum fortunatum*) those were predominant (after pioneer species) across the four forests. In the case of plant functional diversity, it better explained fungal than bacterial diversity. Complex litter biopolymers are decomposed by fungi, and thus, the properties of litter should be reflected in fungi, which might be one of the reasons that plant functional diversity better predicts fungi than bacteria [46]. Again, the greater dependency of fungi on plant products [47] and fungal mycelia from the plant rhizosphere extending to the bulk soil might be the other probable reasons for the better prediction of fungi [46].

In the present study, there was a 3% contribution from soil properties in explaining fungal and bacterial diversity. Recent article reported 2-4% contribution of soil properties in explaining variations of fungal and bacterial diversity in a species-rich grassland [48]. Greater contributions of plant attributes than soil properties in explaining soil microbial diversity were also reported by several empirical studies [6,46,49]. Usually, soil chemical properties, latitudinal distances, or climatic factors (MAP and MAT) are the dominant drivers of soil microbial communities [35,50,51]. However, the climatic factors and soil types of the four forests were nearly uniform in this study, with the only difference being the species that were introduced or planted. This might be the probable reason of plant attributes being the dominant drivers of soil microbial communities other than the environmental drivers. Increase in soil fertility (SOC and TN)through plant-mediated inputs increased soil microbial richness by providing more resources and available niches [13].

Functional traits have shown significant effects on soil microbial populations. The aboveground traits (e.g., SLA, LDMC, VD, LC, LN, LP) and belowground traits (SRL, RDMC, RC, RN, RP) were important predictors of soil microbial communities. Overall, belowground traits better explained soil microbial diversity than aboveground traits. Plant functional traits related to photosynthesis, carbon chemistry of litter and roots, hydraulic conductance and nutrient acquisition can profoundly influence soil microbes [31,32,38,52]. Moreover, the functional traits found to influence the soil microbial communities were considered as the fundamental indicators controlling the quality and quantity of inputs that stimulate soil fertility [38,53]. These functional traits promote niche partitioning and rhizodeposition via the diversity of resources, which in turn influence fungal and bacterial richness. Studies that reported the link between fungi and bacteria richness and diversity with plant functional traits (SLA, LDMC, RDMC, Shoot C, N, root C, N) from grassland and forest ecosystems [31,32,52] were consistent with our findings. A recent article reported functional traits related to nutrient acquisition can better predict fungal and bacterial diversity [31]. In this study, belowground traits were the ones that mostly influenced the variation observed, due to a closer association of soil microbes with roots [27]. Additionally, as root traits determine the quality and quantity of plant carbon and nitrogen supply for the activity of soil microbial communities. At last, plant responses to soil properties that directly or indirectly influence soil microbial communities can be reflected through the root traits.

In summary, the present study showed that plant attributes are fundamental in driving microbial diversity of restored forests. Among the plant attributes, plant taxonomic diversity explained more variation of the fungal and bacterial diversity than plant functional diversity. Specific species were also found to influence fungal and bacterial richness. Plant functional diversity alone (individual effect) only explained of fungal not bacterial diversity. Indeed, our results suggest that experiments studying the influence of plant functional diversity on soil microbial communities should include both above and below ground plant functional traits. Furthermore, fora better understanding of the effect of plant functional diversity on soil microbial communities, plant functional traits of understorey species need to be incorporated.

## 4. Materials and Methods

### 4.1. Experimental Site

The experimental site, the Heshan National Field Research Station of Forest Ecosystem (112°50′ E and 22°34′ N), is located in the subtropical hilly region of Guangdong Province, southern China. The study site is characterized by a subtropical climate with a mean annual temperature of 21.7 °C, a mean annual rainfall of 1700 mm, and a hot and humid rainy season beginning in April and ending in September. The period from October to March is the cool, dry season [54]. The soil of the region is developed from sandstone and is classified as an ultisol [54]. Previously, the site was overexploited and denuded, resulting in severe land degradation. In 1984, an attempt was made to restore the degraded hills through the creation of different plantations. The area restored is approximately 12.22 hectares, with several small patches converted into four forests. Each forest was divided into several small sites based on its orientation and slope positions. The four types of forests include a monoculture of *Acacia mangium*, a mixed forestof *Eucalyptus* species (*E. exserta*, *E. citriodora*, and *E. camaldulensis*) (EE), a mixed forestof coniferous species (*Cunninghamia lanceolata* and *Pinus massoniana*) (MC), and a mixed native species forest (*Schima superba* and *S. wallichii*) (NS). One-year-old healthy saplings were planted at a 2.5 m × 2.5 m spacing in 1984. Anthropogenic activities were prohibited in the forest areas, which allowed the forests to grow naturally.

### 4.2. Field Plots and Plant Sampling

The sampling plots were established in 2017, and a plant inventory was undertaken during July and August at 3 sites (patch) of each forest. There were several small patches of four forests and these patches were independent of each other based on orientation and landscape position. The plots were set up based on the slope position (upper slope, middle slope and lower slope), and at least one plot of 10 m × 10 m size was sampled in each position. For this study, the plant inventory and soil sample collection were conducted in 3 plots in each patch, with 9 plots in each forest. Plant species were divided into three layers (>3 m) and shrub layers (<3 m) during the inventory. Plant composition (richness and diversity) was determined from this inventory. A complete list of species is presented in Table A8.

### 4.3. Plant Functional Traits

Aboveground and belowground plant functional traits were measured following the standard protocols described by Cornelissen et al. [55]. Leaf and root samples were collected from the dominant species. The dominant species were selected based on their relative abundance. We collected plant samples from 40 species from the 4 forests, and some species were common among all forests. We measured maximum plant height, SLA, LDMC, VD, LC, LN, and LP, respectively. The root traits were RD, SRL, RDMC, RC, RN, and RP, respectively. We collected leaf and root samples from five individuals and five samples (leaves and roots) from each individual to measure the plant functional traits. SLA and LDMC were measured from senescent leaves fully exposed to sunlight. Collected leaves were immediately wrapped with tissue paper, sprayed with water and stored in an ice box. The leaf area of fresh and turgid leaves was measured with a leaf area meter (Li-Cor 3100C Area Meter, Li-Cor, Lincoln, NE, USA). The leaves were dried to a constant mass, and SLA was measured by dividing the leaf area by its dry mass; LDMC was measured from the ratio between oven dry mass and water-saturated fresh mass. Leaf vein density was measured following the standard protocol described by Peìrez-Harguindeguy et al. [53], which involves leaf clearing using NaOH-H_2_O (5% w/v) solution for 24–72 h followed by bleaching with 2% w/v NaOCL-H_2_O. Then, the leaves were dehydrated and stained before taking a photograph under a light microscope. The images were processed with ImageJ software to measure the vein density. Fine root samples were collected from the base of the trees by carefully excavating the surface soil with a specially constructed fork to expose the main lateral roots. We gently excavated soils to collect fine roots, and the depth of soil excavation was different for species. Roots were washed with deionized water and stored in a Formalin-Aceto-Alcohol solution (90 mL 50% ethanol, 5 mL 100% glacial acetic acid, 5 mL 37% methanol); another portion was placed on ice and transported to the laboratory within 4 hours. The fine roots were then scanned at 300 dpi with a scanner (EPSON Perfection V850 Pro) to obtain the root images, and image processing software WinRHIZO Pro (Regent Instruments Inc., Sainte-Foy Sillery-Cap-Rouge, QC, Canada) was used to obtain the root length, root diameter, and root volume. Furthermore, the roots were dried to a constant mass; SRL was measured by dividing the root length by its dry mass. RDMC was measured from the ratio between root dry mass and fresh mass. To measure the nutrient content, dried leaf and root samples were ground in a ball mill. The C and N contents in the leaves and roots were determined (Vario elemental analyzer, Langenselbold, Germany); the phosphorus content in leaves and roots was determined by combustion and digestion in sulfuric acid following the molybdenum antimony colorimetric method [32]. We measured the CWM for each trait, which represents the functional diversity [12]. CWM is the abundance-weighted mean trait value for a community and was calculated with the following formula:
CWM (trait_x_) = ΣP_i_T_i_(1)
where P_i_ is the relative abundance for the i^th^ species in the community and T_i_ is the mean trait value of the i^th^ species in the community. We measured the single trait functional dispersion in each forest community [56] using the following formula:(2)$$\mathrm{FD}=\overset{n}{\underset{i=1}{\sum }}p_{i}\frac{\left|T_{i}-CWT_{i}\right|}{\sum_{i}^{n}\left|T_{i}-CWT_{i}\right|}$$

Furthermore, we measured FD including all traits following Laliberté and Legendre [57]:
FD = ∑ (P_ij_Z_j_) / ∑ A_j_(3)
c = ∑ (P_ij_T_jj_) / ∑ A_j_(4)
where T_ij_ is the value of trait i for species j, Aj is the abundance of species j, and c is used to calculate Z_j_, the distance of species j to the weighted centroid.

### 4.4. Soil and Climate

Surface soil samples (0–20 cm) were collected with a stainless-steel soil auger. We randomly collected three soil cores from each plot and then homogenized them into one composite sample. The soil auger was cleaned and sterilized (70% ethanol) properly between each soil sample collection to prevent cross-contamination. After sampling, soils were sieved (2 mm mesh) and divided into two fractions; one fraction was used for chemical analysis, and the other was stored in a −80 °C freezer for molecular analysis. The soil chemical properties determined in the study include SOC, TN, TP, TK, soil moisture content, and soil pH. SOC and TN were determined on an Elementar analyzer (Vario Elemental Analyzer, Langenselbold). TP was measured following a similar method used for plant properties [32], and TK was measured by extraction with 1 M NH_4_OAc, and an atomic absorption flame spectrophotometer was used to determine the TK [58]. The fresh soil samples were oven dried at 105 °C for 24 h, then we determined the soil moisture content gravimetrically. A Delta 320 pH meter (Metler-Toledo Instruments Co., Shanghai, China) was used to determine the soil pH in a soil suspension with a soil:water ratio of 1:2.5 (w/v). Data of MAT and MAP were collected from the weather station at the Heshan National Field Research Station of Forest Ecosystem.

### 4.5. Soil Microbial Community Composition

A Powersoil^®^ DNA Isolation Kit (Mo Bio Laboratories, Carlsbad, CA, USA) was used to extract the soil DNA. Prior to amplification, soil samples were diluted to 1:10. An Illumina MiSeq platform was used to target the ITS2 region of fungi and the 16S rRNA gene of bacteria conducted at GENEWIZ Inc. (Suzhou, China). DNA samples were quantified using a Qubit 2.0Fluorometer (Invitrogen, Carlsbad, CA, USA). Oligonucleotide primers in the ITS 2 region were amplified using forward primers containing the sequence “GTGAATCATCGARTC” and reverse primers containing the sequence “TCCTCCGCTTATTGAT”. The v3 and v4 hypervariable regions of bacteria were amplified using forward primers containing the sequence “CCTACGGRRBGCASCAGKVRVGAAT” and reverse primers containing the sequence “GGACTACNVGGGTWTCTAATCC”. DNA templates (50 ng) were used to generate amplicons using the above-mentioned primers [59,60]. PureLink PCR purification kits (Invitrogen, Paisley, UK) were used for purification of the amplicons that were run on a 3730 DNA analyzer (Applied Biosystems, CA, USA). The 1st round of PCR products was used as templates for 2nd round amplicon enrichment PCR. In addition to the ITS target-specific sequences, the primers also contained adaptor sequences allowing uniform amplification of the library with high complexity ready for downstream NGS sequencing on the Illumina Miseq platform. DNA libraries were validated by an Agilent 2100 Bioanalyzer (Agilent Technologies, Palo Alto, CA, USA). DNA libraries were multiplexed and loaded on an Illumina MiSeq instrument according to the manufacturer’s instructions (Illumina, San Diego, CA, USA). Sequencing was performed using a 2x300/250 paired-end (PE) configuration; image analysis and base calling were conducted by the MiSeq control software (MCS) embedded in the MiSeq instrument. A clustering program (VSEARCH 1.9.6) was used to select operational taxonomic units (OTUs) at 97% sequence similarity. The Ribosomal Database Program (RDP) classifier was used to assign a taxonomic category to all OTUs at the confidence threshold of 0.8. The RDP classifier uses the UNITE ITS database and Silva 132 database as a reference database. Sequenced OTU groups were rarefied before computing diversity indices (Figure A4). The UCHIME algorithm was used to compare the sequence with the reference database, and chimera sequences were removed from sequencing results. The QIIME data analysis package was used to calculate the Shannon diversity index for both bacteria and fungi. The raw reads of fungal ITS and bacterial 16S rRNA gene sequences were uploaded and stored in the Sequence Read Archive (https://submit.ncbi.nlm.nih.gov/subs/sra/) under the Bio project number “PRJNA578999” and “PRJNA578995”.

### 4.6. Statistical Analyses

To identify variations of the plant and microbial communities in the four forests in response to forest restoration, we performed analysis of variance (ANOVA) and post hoc (LSD) tests. Furthermore, we used an analysis of covariance (ANCOVA) test to determine the effect of forest types on the fungal and bacterial richness and diversity. We included CWM and FD values in the model. For CWM values, we created a single variable through principal component analysis. The first PC1 axis described 62.89% variation and was significantly linked with some CWM traits. Subsequently, to test our second hypothesis that plant taxonomic diversity might contribute more in explaining microbial diversity, we first examined the relationship between plant taxonomic diversity and microbial diversity using partial least square regression (PLRS) analysis combining the four forests. For PLRS analysis, we used fungal/bacterial richness as an endogenous (dependent) variable and plant richness as the exogenous (independent) variable. The redundancy analysis (RDA) in CANOCO 4.5 software [61] was also performed to measure the variation of soil microbes (species variable) described by the plant attributes and soil predictors (environmental variables). Moreover, to identify the proportional explanatory power of different predictors (taxonomic diversity, functional diversity, and soil properties) regulating the fungal and bacterial communities, we performed variance partitioning in the vegan package of R [62]. Species richness and plant diversity were grouped together to represent taxonomic diversity. Significant CWM traits and multi trait FD were in the functional diversity group, significant soil properties identified by previous multivariate analysis and Pearson correlation test represented the group of soil predictors. The fungal/bacterial richness and diversity were grouped together as response variables. To test our third hypothesis that root traits explain more variation of soil microbial communities than leaf traits, we first determined the variation of functional diversity (CWM and FD) among the four forests by ANOVA test and followed by the mean separation test. We used the distance-based linear regression model [63] in R to evaluate the proportion of fungal and bacterial community variation explained by the plant traits. We constructed different models using the plant attributes as the predictor, and we selected the best-fitted model that explained fungal and bacterial richness. The best-fitted model was selected according to the Akaike information criterion (AIC). A lower value of the AIC between models represents the best-fitted model. The variance infiltration factor (VIF) was not a concern because it was <2.0 for all models.

## 5. Conclusions

Disentangling of plant-microbe interactions during restoration can improve our understanding of successful restoration trajectories. This article reports the understanding of plant microbial interaction in four different plantation forests in southern China. Plant taxonomic diversity explained better the soil microbial diversity implies the importance of maintaining high species diversity (richness and diversity) in order to maintain high microbial richness. High microbial richness can have a synergistic impact on the success of forest restoration. Forest restoration with leguminous species was associated with fungal richness and diversity. This result indicated that restoration with productive species might have significant impact on the diversity soil microbes and outcomes of the restoration. In addition, the plant functional traits (aboveground and Belowground) contributed to soil microbial community, belowground traits exhibited a stronger effect than aboveground traits. Such information suggests that restored species through their plant mediated inputs can assist in the sustainable management of restored forest ecosystems. Indeed, microbial communities have an important role in ecosystems functioning and the knowledge of plant-microbe interactions of restored forest ecosystem might trigger restoration success around the globe. Future research on the links between plant attributes and soil microbial communities should focus on a wider range scale, including other microbial groups (e.g., archaea, protists, microeukaryotes and other small metazoans), exogenous nutrient deposition, and climate change regimes.

## Figures and Tables

**Figure 1 plants-08-00479-f001:**
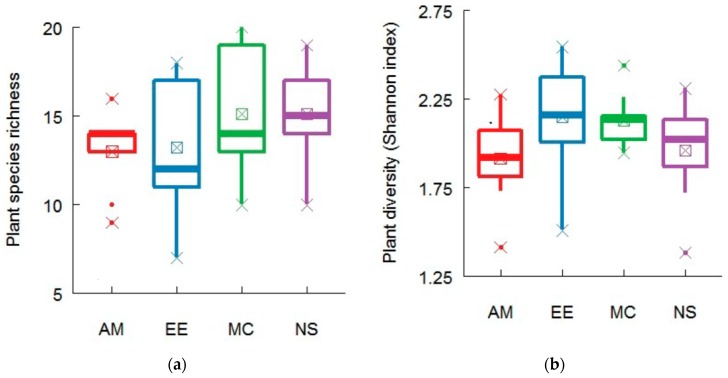
Plant community composition (**a**) plant species richness (number of species) and (**b**) Shannon index of species diversity in different forests of the Heshan forest station, southern China. Letters above the box are the findings from post hoc test. Same letter in the boxes represents statistically similar while different letters represents statistically different. Significance level at *P* < 0.05.

**Figure 2 plants-08-00479-f002:**
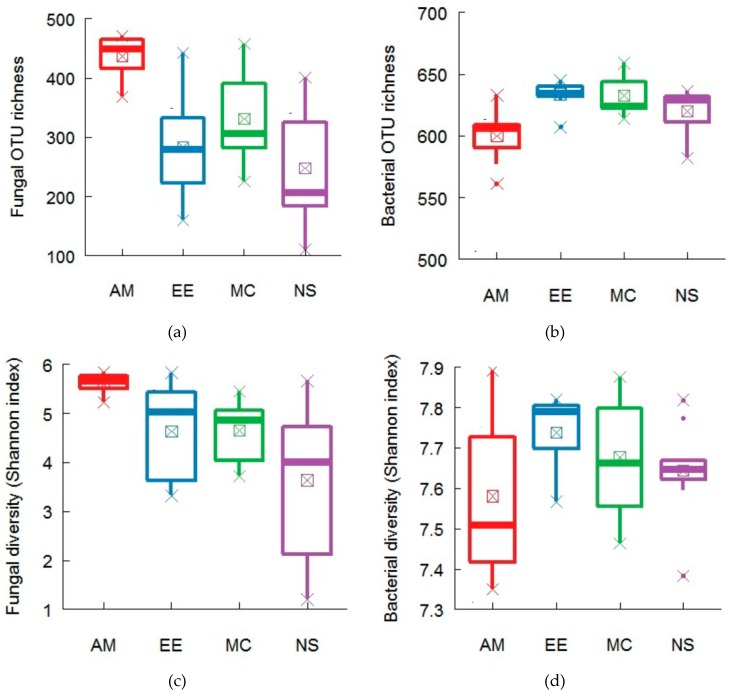
Fungal and bacterial OTU richness and diversity in different forests of the Heshan forest station, southern China. Panel represents (**a**) fungal richness (**b**) bacterial richness (**c**) fungal diversity (**d**) bacterial diversity. P value represents the significance level at *P* < 0.05.

**Figure 3 plants-08-00479-f003:**
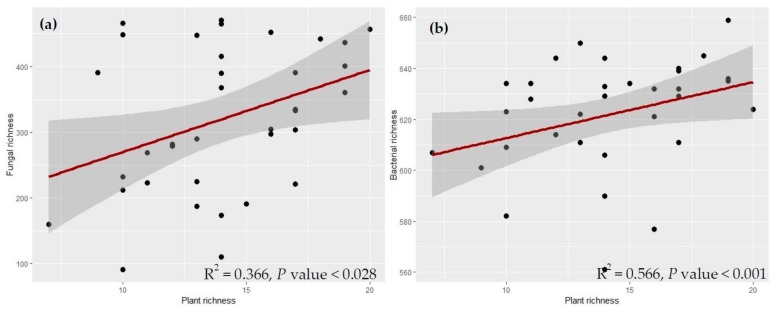
Relationships of species richness with (**a**) fungal and (**b**) bacterial OTU richness. The shaded areas show 95% confidence interval, and the red lines represent the fitted line of the partial linear regression. P value represents the significance level at *P* < 0.05.

**Figure 4 plants-08-00479-f004:**
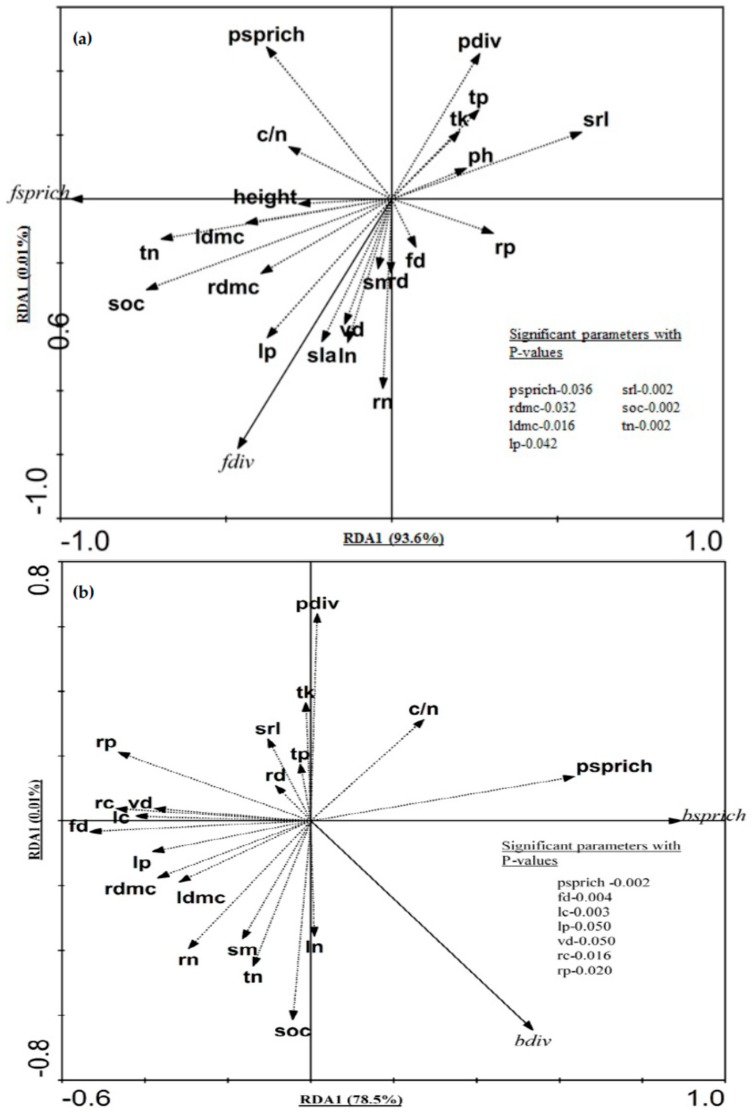
Redundancy analysis (RDA) ordination bi-plot of soil (**a**) fungi, (**b**) bacteria, plant taxonomic diversity (richness and diversity), plant functional traits, and soil and climatic factors. The solid lines indicate the species (fungi and bacteria) variables and the dashed lines indicate the environmental variables (plant attributes, soil and climatic factors). Significant variables are listed inside the plot.

**Figure 5 plants-08-00479-f005:**
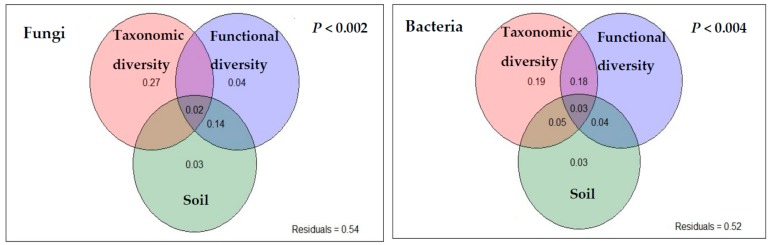
Relative contribution from plant taxonomic diversity, functional diversity, and soil properties to determine belowground fungal and bacterial diversity by variance partitioning analysis (VPA). Significant indicators of each predictors (after RDA) were included. Taxonomic diversity includes both species richness and plant diversity (Shannon index); functional diversity included values of both community-weighted mean (CWM) traits and multi-trait functional dispersion (FD_is_). P value represents the significance level at *P* < 0.05.

**Table 1 plants-08-00479-t001:** Variation in community weighted mean trait (CWM) and multi-traits functional dispersion (FD) across four forests (Data presented in column are mean ± standard error of mean).

Plant Traits	AM	EE	MC	NS	F	*P*
Community weighted mean (CWM) trait						
Height	3.22±0.27	3.28 ± 0.23	3.14 ± 0.23	2.64 ± 0.15	1.66	0.2022
SLA	0.19 ± 0.01a	0.15 ± 0.01b	0.14 ± 0.01b	0.14 ± 0.04b	1.48	**0.0212**
LDMC	261.23 ± 11.23a	197.85 ± 17.09b	234.28 ± 6.01ab	227.86 ± 6.10ab	5.17	**0.0067**
VD	4.50 ± 0.28a	3.98 ± 0.37ab	4.09 ± 0.15a	3.31 ± 0.16b	3.83	**0.0225**
LC	386.66 ± 11.12	333.74 ± 29.26	370.49 ± 8.86	367.49 ± 11.59	1.53	0.2331
LN	20.87 ± 1.23a	22.32 ± 1.95a	21.60 ± 1.18a	16.57 ± 1.25b	3	**0.0504**
LP	1.75 ± 0.08a	1.53 ± 0.13a	1.24 ± 0.05b	0.81 ± 0.04c	23.16	**0.0000**
RD	0.51 ± 0.02	0.45 ± 0.04	0.48 ± 0.02	0.48 ± 0.09	0.25	0.8626
SRL	5.93 ± 0.25b	6.56 ± 0.53ab	5.81 ± 0.19b	7.49 ± 0.20a	6.1	**0.0031**
RDMC	262.00 ± 13.52a	196.68 ± 16.97b	224.92 ± 4.74b	222.11 ± 5.39b	6.19	**0.0029**
RC	378.89 ± 12.35a	305.23 ± 26.62b	354.30 ± 8.01a	349 ± 11.37ab	3.32	**0.0367**
RN	11.67 ± 0.39a	11.20 ± 1.11a	11.01 ± 0.36ab	9.68 ± 0.28b	1.75	**0.0081**
RP	0.56 ± 0.01	0.51 ± 0.04	0.58 ± 0.02	0.59 ± 0.02	1.59	0.2175
Multi-trait Functional diversity						
FD	0.17 ± 0.02a	0.09 ± 0.00b	0.12 ± 0.00b	0.17 ± 0.04a	7.23	**0.0013**

Letters in rows are from post hoc test; same letter in the rows represents statistically similar while different letters represent statistically different. P value showing statistical significance (P<0.05). Height (Maximum plant height), Specific leaf area (SLA), Leaf dry matter content (LDMC), Vein density (VD), Leaf carbon content (LC), Leaf nitrogen content (LN), Leaf phosphorus content (LP), Root diameter (RD), Specific root length (SRL), Root dry matter content (RDMC), Root carbon (RC), Root nitrogen (RN), Root phosphorus (RP).

**Table 2 plants-08-00479-t002:** Prediction of belowground microbial diversity (fungi and bacteria) using the best fitted model.

Soil Microbial Communities	Predictors	Indicators	% Variation Explained	*P*
Fungi	Taxonomic diversity Above ground traits Belowground traits	PSPRICH, Height, SLA, LDMC, VD, LN, LP, SRL, RDMC, RC, RN, RP	0.76	<0.001
	Above ground traits Belowground traits	Height, SLA, LDMC, VD, LN, LP, SRL, RDMC, RC, RN, RP	0.68	<0.001
	Above ground traits	Height, LDMC, VD, LN, LP	0.42	<0.001
	Belowground traits	SRL, RDMC, RC, RP	0.43	<0.001
Bacteria	Taxonomic diversity Above ground traits Belowground traits	PSPRICH, LDMC, VD, LC, LN, LP, RDMC, RN, RP	0.45	<0.001
	Above ground traits Belowground traits	LDMC, VD, LC, LN, LP, RDMC, RN, RP	0.31	<0.011
	Above ground traits	LDMC, LC, LN, LP	0.16	<0.032
	Belowground traits	RDMC, RC, RN, RP	0.22	<0.036

Taxonomic diversity and plant functional traits were used as predictor variables and fungal/bacterial richness was used as explanatory variables. P value showing statistical significance (*P* < 0.05). PSPRICH (Plant species richness); SLA (Specific leaf area); LDMC (Leaf dry matter content); VD (Vein density); LC (Leaf carbon content); LN (Leaf nitrogen content); LP (Leaf phosphorus content); SRL (Specific root length); RDMC (Root dry matter content); RC (Root carbon content); RN (Root nitrogen content); RP (Root phosphorus content).

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
