# Peer review of "Plant Taxonomic Diversity Better Explains Soil Fungal and Bacterial Diversity than Functional Diversity in Restored Forest Ecosystems"

_plants, 2019, doi:10.3390/plants8110479_

Round 1

Reviewer 1 Report

This is a well designed study that sheds light on soil-plant-microbe interactions. The authors are encouraged to consider the role that obligate relationships may influence species diversity and how this may account for the positive correlation between plant species richness and soil microbe diversity.

Reviewer 2 Report

The purpose of the study was to investigate the role of plant diversity (taxonomic and functional diversity) and plant functional traits (aboveground and belowground) as predictors of belowground soil fungal and bacterial diversities in four subtropical plantation forests of southern China. For that a trait‐based approaches for plant properties and high‐throughput Illumina sequencing for fungal and bacterial diversity were applied. The authors manage to gather a very thorough and extended dataset of microbial and environmental variables that were used to draw conclusions based on appropriate statistical analysis. The article is well written and concise, although at some points it needs some re-arrangements to read less scatter. Therefore, I recommend the article to be published after minor revision to address the comments below.

General comments:

Abstract: reads to scatter. Lacking connection words.

Introduction: Keep all the hypotheses together at the end of the introduction, after the description of the main objectives.

Figure 3: delete the – after P-value in figure 3-A. Change the figure legend to read: “Relationships of plant species richness with (a) fungal and (b) bacterial OTU richness.”

Material and Methods: Is the sequencing available in a public repository?

Specific comments

Line 23: missing a space after the full stop.

Line 41: remove the word “of”.

Lines 45-47: Modify to read: “Vegetation types having contrasting diversity of plant communities will have a distinct effect on soil properties through plant inputs, which in turn has diverse effects on soil microbial communities.”

Lines 61-63: Modify to read: “Plant taxonomic and functional diversity affect soil microbial diversity during restoration by altering the available resources niche differentiation and resource partitioning [5,12-16].”

Lines 84-87: This sentence seems repetitive, I advise to delete.

Lines 118-123: Please keep the order of the description as shown in the figure, or alter the order of the graphs in Figure 1.

Line 136: Add a the after “among”.

Line 247: The solid lines indicate the species variables for bacteria and fungi, right? Not only fungi as described in the paper.

Lines 326-330: Sentence too long. Suggestion: “Moreover, the multi‐trait FD and single trait functional diversity measures were maximum in the AM forest, which indicates that a higher resource availability (nutrients entering the soil via the plant parts) was present in this site, leading to more availability of niche space for fungi [39,40,41,42].”

Lines 333-335: Change to read: “The quality of nutrients and lower plant diversity in the AM forest compared to the other forests might be attributed to the lower bacterial richness and diversity. In contrast, bacterial richness was higher in EE plantation…”

Line 338: Authors need to introduce references for these statements.

Line 374: Lacks references.

Lines 378-379: Modify to read: “Studies that reported the link between fungi and bacteria diversity and richness with plant functional traits…”

Lines 380-386: Needs some connecting words. Suggestion: “A recent article reported that functional traits related to nutrient acquisition can better predict fungal and bacterial diversity [31]. In this study, belowground traits were the ones that mostly influenced the variation observed, due to a closer association of soil microbes with the roots [27]. Additionally, root traits determine the quality and quantity of plant carbon and nitrogen supply for the activity of soil microbial communities. At last, plant responses to soil properties that directly or indirectly influence soil microbial communities can be reflected through root traits.”

Line 409: change the word “bands”. Maybe you mean plots, areas?!

Line 421: the 40 species were collected in total for all the sites or at each site?

Lines 426-427: Remove sentence due to repetition.

Lines 472-473: Change to read: “one fraction was used for chemical analysis, and the other was stored at −80 °C for molecular analysis”.

Lines 490-495: The primers used were taken from the literature? Then the description lacks references for it.

Line 497: Is this second PCR an index PCR to add the tags or a third PCR was performed? Please explain.

Line 511: Change the word “chemical” to chimera.

Lines 545-551: Modify to read: “Forests with leguminous plant were associated with fungal richness and diversity. Plant taxonomic diversity explained better the soil microbial diversity than functional diversity. Whereas, the plant functional traits (aboveground and belowground) contributed for soil microbial community. Belowground traits exhibited a stronger effect than aboveground traits. Future research on the links between plant attributes and soil microbial communities should focus on a wider range scale, including other microbial groups (e.g., archaea, protists, microeukaryotes and other small metazoans), exogenous nutrient deposition, and climate change regimes.”

Reviewer 3 Report

This study investigates the fungal and microbial diversity in forest plantations of different tree species composition. Overall, the paper is clearly written and the analytical methods seem appropriate, but the paper would benefit from some clarifications. Most importantly, the conclusions remain unclear–the conclusions paragraph merely repeats the main results, but does not say anything about the significance of them, how these results could/should be considered in forest restoration? No suggestions or any advice is given, thus, giving the impression that the results are of no use and the study was insignificant. This part could and should be improved a lot. Other comments:

Use consistent terminology, for example the legend of Table 1 says "four forest ecosystems", Table A4 says "four plantations" and Table A5 says "four forests". Decide whether these are plantations, forests or ecosystems, and use the same terminology throughout the text. In the same manner, the species richness, taxonomic diversity and richness are used unclearly, probably all meaning the same thing.

Title: the title is confusing; plant taxonomic diversity... (what that even means, apparently species richness?) ...better explains... (better than what??)

Abstract (and elsewhere): richness apparently means number of species, so please talk about species richness or number of species to be clear. Here and there is used term "taxonomic diversity", what does that mean? Be clear and constant with the terms, explain them when first used and use the same term throughout the paper to describe the same thing.

Results: explain the forest types (EE, MS, AM) (and other abbreviations likewise) when you use them for the first time. Obviously the journal style of having results before the methods unnecessarily complicates the matters, but nevertheless, you should write the text so that it is possible to understand the text without the need to jump to the methods section to check what results are explained. This same comment applies to the abbreviated plant attributes and soil properties (line 196).

Discussion: line 316 & 327 replace "maximum" with "highest". Lines 335-336 "bacterial richness was higher in EE plantations..." do you mean highest? If higher, then higher compared to what? Since this is among the studied forests, you probably mean highest? Line 343 "supports our second hypothesis" What is this second hypothesis? And what is the first hypothesis? Either write out what is supported, or clearly write out your hypothesis at the beginning of the discussion (and introduction). Line 344 "with increasing species richness" probably means plant species richness, right? Clarify. Line 348-350 the sentence beginning with "In our study..." needs to be rewritten, its meaning remains unclear at the moment. Line 363-365 the same problem; the sentence makes currently no sense. Line 377 what does the rhizodeposition mean?? It would be nice to have some concluding remarks at the end on discussion, the format of having discussion first, materials and methods after that and conclusions at the end is very unfortunate, but currently there are no real conclusions anywhere, leaving the importance and relevance of this study completely open. 

Materials and methods: it remains a bit unclear how the sampling was done. E.g. the sentence at lines 409-410 does not help. Line 515 "to test our first hypothesis" and line 521 "to test our second hypothesis" clarify what these hypotheses are! Line 421-423, it remains unclear how the dominant species were selected.

Conclusions: these do not really conclude anything, but only summarizes the results already summarized in the abstract. The sentence "Forest with leguminous plant is more associated with fungal richness and diversity" does not really mean anything.

Round 2

Reviewer 3 Report

Manuscript is now much clearer, good work.